# Practical Structured Riemannian Optimization with Momentum by using Generalized Normal Coordinates

**Wu Lin**                                                                      WLIN2018@CS.UBC.CA
**Valentin Duruisseaux**                                                          VDURUISS@UCSD.EDU
**Melvin Leok**                                                                     MLEOK@UCSD.EDU
**Frank Nielsen**                                                          FRANK.NIELSEN@ACM.ORG
**Emtiyaz Khan**                                                          EMTIYAZ.KHAN@RIKEN.JP
**Mark Schmidt**                                                              SCHMIDTM@CS.UBC.CA

**Editors:** Sophia Sanborn, Christian Shewmake, Simone Azeglio, Arianna Di Bernardo, Nina Miolane

## Abstract

Adding momentum into Riemannian optimization is computationally challenging due to the intractable ODEs needed to define the exponential and parallel transport maps. We address these issues for Gaussian Fisher-Rao manifolds by proposing new local coordinates to exploit sparse structures and efficiently approximate the ODEs, which results in a numerically stable update scheme. Our approach extends the structured natural-gradient descent method of Lin et al. (2021a) by incorporating momentum into it and scaling the method for large-scale applications arising in numerical optimization and deep learning.

**Keywords:** Riemannian Manifolds, Matrix Lie Groups, Numerical Optimization

## 1. Introduction and Background

Riemannian gradient methods are useful for optimization on manifolds. Momentum (Goodfellow et al., 2016) is a common approach to improve gradient-based methods. Unfortunately, adding momentum into Riemannian optimization is computationally intensive due to the intractable ODEs needed to define the exponential and parallel transport map. Moreover, Riemannian methods could perform poorly and be numerically unstable in single-precision floating-point settings such as training large-scale neural networks. Straightforward approximations of the ODEs in global coordinates involve computing nonsparse and high-order derivatives which limits the large-scale usage of Riemannian methods.

Many well-known algorithms in machine learning can be derived from Gaussian Fisher-Rao manifolds, for example, see Khan and Rue (2021); Lin et al. (2021a). In this work, we propose special local coordinates for Gaussian Fisher-Rao manifolds to relax the requirement of the use of global coordinates, and the exact exponential and Riemannian transport maps required by previous methods, and exploit sparse Lie-group structures to obtain numerically stable updates. It is challenging to add momentum in local coordinates, which involves the computation of the metric, the exponential and transport maps, and the change of local coordinates. Our Riemannian momentum update can be viewed as Euclidean GD with momentum in a special coordinate. We use the approach to extend the structured natural-gradient descent (NGD) method (Lin et al., 2021a) by adding momentum to the method, and scaling it to large-scale applications (see Fig.1). We will begin with the basics, introduce the method, and conclude with an example. Additional results can be found in appendices.

### 1.1. Riemannian Gradient Descent and Natural-gradient Descent

Consider a geodesically complete manifold $\mathcal{M}$ with the Fisher-Rao $\mathbf{F}$ metric. The metric is known as the Fisher information matrix and is defined in A.2. Under a global parametrization $\boldsymbol{\tau}$, Riemannian gradient descent (Bonnabel, 2013) (RGD) is defined as

$$\boldsymbol{\tau} \leftarrow \text{RExp}(\boldsymbol{\tau}, -\alpha\mathbf{F}_{\tau}^{-1}(\boldsymbol{\tau})\mathbf{g}(\boldsymbol{\tau})), \tag{1}$$

where $\mathbf{v} := \mathbf{F}_\tau^{-1}(\boldsymbol{\tau})\mathbf{g}(\boldsymbol{\tau})$ is a Riemannian gradient, $\mathbf{g}(\boldsymbol{\tau})$ is a Euclidean gradient (1-form), $\mathbf{F}_\tau$ is the metric, $\mathrm{RExp}(\boldsymbol{\tau}, \mathbf{v})$ is the Riemannian exponential map (defined in A.4) induced by the geodesic ODE at the current $\boldsymbol{\tau}$ with Riemannian direction $\mathbf{v}$, and $\alpha$ is a step-size. The exponential map often does not admit a closed-form expression (see Table 1). For an approximation of the exponential map, we have to evaluate the Christoffel symbols $\Gamma_{cb}^a(\boldsymbol{\tau})$ defined at A.3, which involves computing partial derivatives of the Fisher matrix.

Natural-gradient descent (NGD) is a special case of RGD in (1) that uses a first-order Taylor approximation of the exponential map to avoid evaluating the Christoffel symbols,

$$\boldsymbol{\tau} \leftarrow \boldsymbol{\tau} - \alpha\mathbf{F}_\tau^{-1}(\boldsymbol{\tau})\mathbf{g}(\boldsymbol{\tau}) = \mathrm{Ret}(\boldsymbol{\tau}, -\alpha\mathbf{F}_\tau^{-1}(\boldsymbol{\tau})\mathbf{g}(\boldsymbol{\tau})), \tag{2}$$

This approximation, which is known as the Euclidean retraction, is defined as,

$$\mathrm{RExp}(\mathbf{x}, \boldsymbol{\nu}) = \mathbf{r}(1) \approx \mathrm{Ret}(\mathbf{x}, \boldsymbol{\nu}) := \mathbf{r}(0) + \dot{\mathbf{r}}(0)(1-0) = \mathbf{x} + \boldsymbol{\nu},$$

where $\mathbf{r}(t)$ is the solution of the geodesic ODE with initial values $\mathbf{r}(0) = \mathbf{x}$ and $\dot{\mathbf{r}}(0) = \boldsymbol{\nu}$.

## 1.2. Standard Riemannian Normal Parametrization

We now discuss updates using a special local parametrization that will allow us to reduce computational costs, and we will generalize this later.

Let $\boldsymbol{\tau}$ be the global coordinate, and $\boldsymbol{\eta}$ be a normal (local) coordinate at the current point $\boldsymbol{\tau}^{(\mathrm{cur})}$ so that $\boldsymbol{\tau} = \mathrm{RExp}(\boldsymbol{\tau}^{(\mathrm{cur})}, \mathbf{F}_\tau^{-1/2}(\boldsymbol{\tau}^{(\mathrm{cur})})\boldsymbol{\eta})$, where $\boldsymbol{\tau}^{(\mathrm{cur})}$ and $\mathbf{F}_\tau^{-1/2}(\boldsymbol{\tau}^{(\mathrm{cur})})$ are constants in this coordinate. In the coordinate $\boldsymbol{\eta}$, the origin $\boldsymbol{\eta}_0 := \mathbf{0}$ represents point $\boldsymbol{\tau}^{(\mathrm{cur})}$.

The following properties hold at point $\boldsymbol{\tau}^{(\mathrm{cur})}$ in local coordinate $\boldsymbol{\eta}$ at $\boldsymbol{\eta}_0 = \mathbf{0}$.

$$\boldsymbol{\tau}^{(\mathrm{cur})} = \mathrm{RExp}(\boldsymbol{\tau}^{(\mathrm{cur})}, \mathbf{F}_\tau^{-1/2}(\boldsymbol{\tau}^{(\mathrm{cur})})\boldsymbol{\eta}_0); \ \ \mathbf{F}_\eta(\boldsymbol{\eta}_0) = \mathbf{I}; \ \ \Gamma_{cb}^a(\boldsymbol{\eta}_0) = 0. \tag{3}$$

The update of RGD in Eq. 1 can be written as a NGD update in local coordinate $\boldsymbol{\eta}$:

$$\left.\begin{aligned}
\boldsymbol{\eta}_1 &\leftarrow \boldsymbol{\eta}_0 - \alpha\mathbf{g}(\boldsymbol{\eta}_0), \\
\boldsymbol{\tau}^{(\mathrm{new})} &\leftarrow \mathrm{RExp}(\boldsymbol{\tau}^{(\mathrm{cur})}, \mathbf{F}_\tau^{-1/2}(\boldsymbol{\tau}^{(\mathrm{cur})})\boldsymbol{\eta}_1),
\end{aligned}\right\} \equiv \boldsymbol{\tau} \leftarrow \mathrm{RExp}(\boldsymbol{\tau}, -\alpha\mathbf{F}_\tau^{-1}(\boldsymbol{\tau})\mathbf{g}(\boldsymbol{\tau})), \tag{4}$$

where $\boldsymbol{\eta}_0 = \mathbf{0}$, $\mathbf{F}_\eta(\boldsymbol{\eta}_0) = \mathbf{I}$ and $\mathbf{g}(\boldsymbol{\eta}_0) = \mathbf{F}_\tau^{-1/2}(\boldsymbol{\tau}^{(\mathrm{cur})})\mathbf{g}(\boldsymbol{\tau}^{(\mathrm{cur})})$ is a Euclidean gradient. This update remains on the manifold while NGD in global coordinate $\boldsymbol{\tau}$ (see Eq.2) does not, especially when $\boldsymbol{\tau}$ is in a constraint set (e.g., the positive-definite constraint of $\mathbf{S}$ in Eq.10).

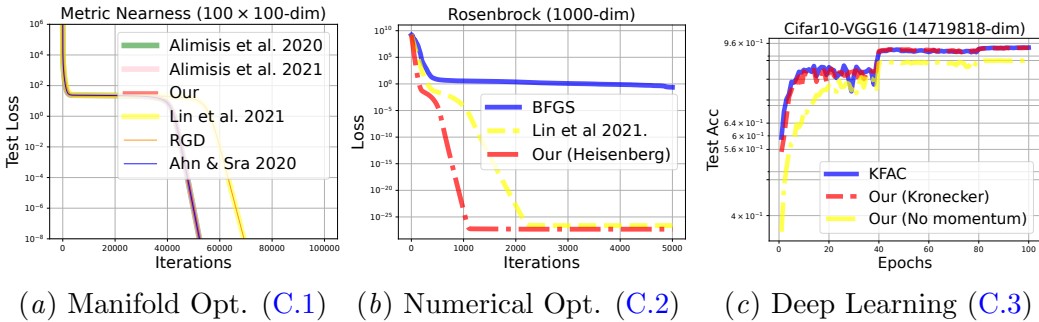

(a) Manifold Opt. (C.1)    (b) Numerical Opt. (C.2)    (c) Deep Learning (C.3)

Figure 1: Performance of various algorithms for 1(a) a positive-definite matrix manifold, 1(b) unconstrained optimization, 1(c) a CNN model. (See App. C for the description)

## 2. Natural Gradient Descent with Momentum

### 2.1. Momentum in Riemannian Gradient Descent

To incorporate momentum in RGD, we have to use a parallel transport map to move a (gradient) vector at the current point $\boldsymbol{\tau}^{(\text{cur})}$ to a new point $\boldsymbol{\tau}^{(\text{new})}$ on a curved space. This map is induced by the transport ODE (defined in Eq.15 and Eq.17 in the appendix). We use the transport map $T_{\tau^{(\text{cur})} \to \tau^{(\text{new})}}$ (defined in A.6) to transport a Euclidean gradient at $\tau^{(\text{cur})}$ to a Euclidean gradient at $\tau^{(\text{new})}$. We consider the following update with momentum weight $\beta$ under global coordinate $\boldsymbol{\tau}$.

$$
\begin{aligned}
\mathbf{m}^{(\text{cur})} &\leftarrow \beta \mathbf{w}^{(\text{cur})} + \alpha \mathbf{g}(\boldsymbol{\tau}^{(\text{cur})}), \\
\boldsymbol{\tau}^{(\text{new})} &\leftarrow \text{RExp}(\boldsymbol{\tau}^{(\text{cur})}, -\mathbf{F}_\tau^{-1}(\boldsymbol{\tau}^{(\text{cur})})\mathbf{m}^{(\text{cur})}), \\
\mathbf{w}^{(\text{new})} &\leftarrow T_{\tau^{(\text{cur})} \to \tau^{(\text{new})}}(\mathbf{m}^{(\text{cur})}),
\end{aligned}
\tag{5}
$$

where $\mathbf{w}$ is a Euclidean gradient and $\mathbf{m}$ is a momentum for Euclidean gradients.

This update is equivalent to the update of Alimisis et al. (2020), shown in A.8, if the exact exponential and the exact transport are used. Many Riemannian momentum methods including Alimisis et al. (2020) use the exact transport map $\hat{T}$ for Riemannian gradients (see A.5) while we use the transport map for Euclidean gradients. In most Gaussian cases, the exact transport maps (see Table 1) are unknown. We choose to approximate the Euclidean transport map $T$ since it is more computationally efficient than the Riemannian map $\hat{T}$ as we will discuss in Sec. 2.3. Thus, our update is distinct from Alimisis et al. (2020).

### 2.2. Generalized Riemannian Normal Parametrizations

Our idea is to use a generalized Riemannian normal parametrization to simplify the computation. The standard Riemannian normal parametrization is computationally challenging since it involves the computation of the Riemannian exponential map.

To get rid of that dependence on the map, we define a generalized Riemannian normal coordinate $\boldsymbol{\eta}$ associated to point $\boldsymbol{\tau}^{(\text{cur})}$ via a diffeomorphic map $\boldsymbol{\tau} = \boldsymbol{\phi}_{\tau^{(\text{cur})}}(\boldsymbol{\eta})$ such that the following conditions hold, where the Christoffel symbols $\Gamma^a_{cb}$ need not vanish at $\boldsymbol{\eta}_0 = \mathbf{0}$.

$$
\boldsymbol{\tau}^{(\text{cur})} = \boldsymbol{\phi}_{\tau^{(\text{cur})}}(\boldsymbol{\eta}_0); \quad \mathbf{F}_\eta(\boldsymbol{\eta}_0) = \mathbf{I}.
\tag{6}
$$

We recover the standard normal coordinate if the exponential map is used as the diffeomorphic map. However, we can instead choose a more efficient map. Moreover, if necessary, we can efficiently compute the Christoffel symbols at $\boldsymbol{\eta}_0$ for Gaussian manifolds.

### 2.3. NGD with Momentum using Generalized Normal Coordinates

We propose an approximation of Eq. 5 by local coordinate $\boldsymbol{\eta}$ associated to global $\boldsymbol{\tau}^{(\text{cur})}$

$$
\begin{aligned}
\mathbf{m}^{(\eta_0)} &\leftarrow \beta \mathbf{w}^{(\eta_0)} + \alpha \mathbf{g}(\boldsymbol{\eta}_0), \\
\left.\begin{aligned}
\boldsymbol{\eta}_1 &\leftarrow \boldsymbol{\eta}_0 - \mathbf{F}_\eta^{-1}(\boldsymbol{\eta}_0)\mathbf{m}^{(\eta_0)}, \\
\boldsymbol{\tau}^{(\text{new})} &\leftarrow \boldsymbol{\phi}_{\tau^{(\text{cur})}}(\boldsymbol{\eta}_1),
\end{aligned}\right\} &\overset{\text{see Eq. } 4}{\approx} \boldsymbol{\tau}^{(\text{new})} \leftarrow \text{RExp}(\boldsymbol{\tau}^{(\text{cur})}, -\mathbf{F}_\tau^{-1}(\boldsymbol{\tau}^{(\text{cur})})\mathbf{m}^{(\text{cur})}), \\
\mathbf{w}^{(\eta_1)} &\leftarrow T_{\eta_0 \to \eta_1}(\mathbf{m}^{(\eta_0)}),
\end{aligned}
$$

where $\mathbf{F}_\eta^{-1}(\boldsymbol{\eta}_0) = \mathbf{I}$, $\boldsymbol{\eta}_0 = \mathbf{0}$, and, $\boldsymbol{\eta}_0$ and $\boldsymbol{\eta}_1$ represent $\boldsymbol{\tau}^{(\text{cur})}$ and $\boldsymbol{\tau}^{(\text{new})}$, respectively.

To use normal coordinate $\boldsymbol{\xi}$ associated to $\boldsymbol{\tau}^{(\mathrm{new})}$ at the next iteration, we have to transform the Euclidean gradient $\mathbf{w}^{(\eta_1)}$ in coordinate $\boldsymbol{\eta}$ into coordinate $\boldsymbol{\xi}$ via the chain rule:

$$\mathbf{w}^{(\xi_0)} = \mathbf{J}(\boldsymbol{\xi}_0)\mathbf{w}^{(\eta_1)}; \;\; \mathbf{J}(\boldsymbol{\xi}) := \frac{\partial \boldsymbol{\eta}}{\partial \boldsymbol{\xi}},$$

where $\boldsymbol{\xi}_0 = \mathbf{0}$, $\mathbf{J}(\boldsymbol{\xi})$ is the Jacobian, and both $\boldsymbol{\xi}_0$ and $\boldsymbol{\eta}_1$ represent $\boldsymbol{\tau}^{(\mathrm{new})}$.

We can compute the Jacobian as $\mathbf{J}(\boldsymbol{\xi}) = \frac{\partial \boldsymbol{\phi}_{\tau^{(\mathrm{cur})}}^{-1} \circ \boldsymbol{\phi}_{\tau^{(\mathrm{new})}}(\boldsymbol{\xi})}{\partial \boldsymbol{\xi}}$ since by construction we have $\boldsymbol{\tau} = \boldsymbol{\phi}_{\tau^{(\mathrm{cur})}}(\boldsymbol{\eta}) = \boldsymbol{\phi}_{\tau^{(\mathrm{new})}}(\boldsymbol{\xi})$. In practice, we could also directly compute $\mathbf{w}^{(\xi_0)}$ by automatic-differentiation. As mentioned in Sec. 2.1, we use the transport map for Euclidean gradients so that the Jacobian can be easy to compute at $\boldsymbol{\xi}_0 = \mathbf{0}$. If the transport map for Riemannian gradients is used, we have to compute the inverse of the Jacobian.

We now discuss approximations of the transport map for Euclidean gradients.

## 2.4. Approximations of the Transport Map

To avoid evaluating the Christoffel symbols, we could use a zero-order Taylor expansion to approximate the parallel transport map as shown below,

$$T_{\eta_0 \to \eta_1}(\mathbf{m}^{(\eta_0)}) = \boldsymbol{\omega}(1) \approx \boldsymbol{\omega}(0) = \mathbf{m}^{(\eta_0)}, \tag{7}$$

where $\boldsymbol{\omega}(t)$ is the solution of the transport ODE with initial value $\boldsymbol{\omega}(0) = \mathbf{m}^{(\eta_0)}$. A more accurate approximation is a first-order approximation as in A.7, where we have to compute the Christoffel symbols at $\boldsymbol{\eta}_0 = \mathbf{0}$ to compute the first-order term $\dot{\boldsymbol{\omega}}(0)$. For Gaussian manifolds, we can simplify the computation since $\boldsymbol{\eta}$ is a generalized normal coordinate.

If we use the standard normal coordinate, the zero-order approximation becomes the first-order approximation since the Christoffel symbols vanish. Our update with the zero-order approximation is exactly the Euclidean momentum update in the local coordinate.

Our goal is to replace the standard normal coordinate with a generalized one. In the next section, we give an example of Gaussian manifolds, where the zero-order approximation is also a first-order approximation since the first-order term $\dot{\boldsymbol{\omega}}(0)$ vanishes. In structured cases, the zero-order approximation works very well for many machine-learning problems as supported by the experiments even when the first order term does not vanish. The first-order approximation does not improve the performance while increasing the computational cost. The first-order term is $O(\alpha^2)$ (See Appendix B), which is small when step-size $\alpha$ is small, especially in noisy, large-scale, or low-precision floating-point settings.

## 3. Gaussian Fisher-Rao Manifolds

We will now illustrate our update described in Sec 2.3 by considering an example of Gaussian Fisher-Rao manifolds. Gaussian Fisher-Rao manifolds play an important role in machine learning. Consider the following unconstrained optimization problem.

$$\min_{\mu \in \mathcal{R}^d} \ell(\boldsymbol{\mu}), \tag{8}$$

where $\ell$ is a real-valued loss function to be minimized. Khan et al. (2017, 2018); Lin et al. (2021a,b) reformulate the problem in Eq. 8 as a new optimization problem over a Gaussian (surrogate) distribution $q(\mathbf{z}|\boldsymbol{\tau})$ with mean $\boldsymbol{\mu}$ and inverse covariance $\mathbf{S}$,

$$E_{q(z|\tau)}[\ell(\mathbf{z})] + E_{q(z|\tau)}[\log q(\mathbf{z}|\boldsymbol{\tau})], \tag{9}$$

with global coordinate $\boldsymbol{\tau} = \{\boldsymbol{\mu}, \mathbf{S}\}$, and where $\ell(\mathbf{z})$ is the loss function defined in Eq. 8.

Khan et al. (2017, 2018) show a connection between NGD (see Eq.2) in global coordinate $\boldsymbol{\tau}$ for problem 8 and Newton's method for problem 9 at steps-size $\alpha = 1$ as

$$\boldsymbol{\mu} \leftarrow \boldsymbol{\mu} - \alpha \mathbf{S}^{-1} E_q[\nabla_z \ell(\mathbf{z})], \quad \mathbf{S} \leftarrow (1-\alpha)\mathbf{S} + \alpha E_q[\nabla_z^2 \ell(\mathbf{z})] \tag{10}$$

This update recovers Newton's method when the expectations are approximated at the mean $\boldsymbol{\mu}$. However, the parameter constraint in $\mathbf{S}$ is not satisfied in Eq.10. Lin et al. (2021a,b) propose NGD updates via local coordinates for structured $\mathbf{S}$ and handle the constraint in $\mathbf{S}$, which enables usage of these Newton-like methods in non-convex and stochastic settings.

We will extend the local coordinates in the method of Lin et al. (2021a,b) by using the generalized normal coordinates. Our extension allows the method to incorporate momentum and improves its performance for large-scale applications. Due to the space limit, we only consider full Gaussians with a constant mean. A structured case is given in Appendix B.

Consider a $d$-dimension Gaussian family with zero mean $q(\mathbf{z}|\boldsymbol{\tau})$, where the global coordinate $\boldsymbol{\tau} = \mathbf{S}$ is the inverse covariance, which is symmetric positive-definite. For efficiency, we propose a new re-parameterization instead of the one in Lin et al. (2021a).

We introduce an auxiliary parameterization $\mathbf{B}$ such that $\boldsymbol{\tau} = \mathbf{B}^T\mathbf{B}$, where $\mathbf{B} \in \mathrm{GL}^{d \times d}$ is an invertible matrix. The auxiliary parameter $\mathbf{B}$ has an additional Lie group structure since it is a member of the general linear group.

We define a local parameterization $\boldsymbol{\eta}$ associated to $\mathbf{B}^{(\mathrm{cur})}$ as $\mathbf{B} = \mathbf{h}(\frac{1}{\sqrt{2}}\boldsymbol{\eta})\mathbf{B}^{(\mathrm{cur})}$, where $\mathbf{h}(\mathbf{N}) = \mathrm{Expm}(\mathbf{N}) := \sum_{n=0}^{\infty} \frac{\mathbf{N}^n}{n!}$ is the matrix exponential (the Lie-group exponential map) and $\boldsymbol{\eta}$ is a symmetric matrix. In practice, we approximate Expm() by a truncation since the exact map is numerically unstable and computationally intensive in noisy, large-scale, and low precision floating-point settings. Local parameter $\boldsymbol{\eta}$ lives in a subspace of the Lie algebra of $\mathbf{B}$ since $\boldsymbol{\eta}$ is symmetric. Note that $\mathbf{m}$ is also symmetric in our update.

In this example, $\boldsymbol{\eta}$ is a generalized Riemannian normal coordinate associated to $\boldsymbol{\tau}^{(\mathrm{cur})}$ via diffeomorphic map $\boldsymbol{\tau} = \boldsymbol{\phi}_{\boldsymbol{\tau}^{(\mathrm{cur})}}(\boldsymbol{\eta}) := (\mathbf{B}^{(\mathrm{cur})})^T \mathbf{h}^T(\frac{1}{\sqrt{2}}\boldsymbol{\eta})\mathbf{h}(\frac{1}{\sqrt{2}}\boldsymbol{\eta})\mathbf{B}^{(\mathrm{cur})}$, where $\boldsymbol{\tau}^{(\mathrm{cur})} = (\mathbf{B}^{(\mathrm{cur})})^T\mathbf{B}^{(\mathrm{cur})} = \boldsymbol{\phi}_{\boldsymbol{\tau}^{(\mathrm{cur})}}(\boldsymbol{\eta}_0)$ and $\mathbf{F}_{\boldsymbol{\eta}}(\boldsymbol{\eta}_0) = \mathbf{I}$ at $\boldsymbol{\eta}_0 = \mathbf{0}$. The Christoffel symbols at $\boldsymbol{\eta}_0$ do not vanish. Thus, $\boldsymbol{\eta}$ is not the standard normal coordinate. We can write $\mathbf{w}$ as $\mathbf{w} = \mathbf{w}^{(\xi_0)} = \mathbf{w}^{(\eta_1)}$ since $\mathbf{w}^{(\xi_0)} = \mathbf{J}(\boldsymbol{\xi}_0)\mathbf{w}^{(\eta_1)} = \mathbf{w}^{(\eta_1)}$ as shown in B.

Our proposed update can be simplified as below, where we can efficiently compute the Euclidean gradient $\mathbf{g}(\boldsymbol{\eta}_0)$ and use the zero-order approximation for the transport map. Although the Christoffel symbols do not vanish, the first-order term of the transport map does vanish as shown in A.7. Thus, this approximation is also a first-order approximation.

$$
\begin{aligned}
\mathbf{m} &\leftarrow \beta\mathbf{w} + \alpha\mathbf{g}(\boldsymbol{\eta}_0), \\
\boldsymbol{\eta}_1 &\leftarrow \boldsymbol{\eta}_0 - \mathbf{m}, \\
\left.\begin{aligned}
\mathbf{B}^{(\mathrm{new})} &\leftarrow \mathbf{h}(\tfrac{1}{\sqrt{2}}\boldsymbol{\eta}_1)\mathbf{B}^{(\mathrm{cur})}, \\
\boldsymbol{\tau}^{(\mathrm{new})} &\leftarrow (\mathbf{B}^{(\mathrm{new})})^T\mathbf{B}^{(\mathrm{new})},
\end{aligned}\right\} &\equiv \boldsymbol{\tau}^{(\mathrm{new})} \leftarrow \boldsymbol{\phi}_{\boldsymbol{\tau}^{(\mathrm{cur})}}(\boldsymbol{\eta}_1), \\
\mathbf{w} &\leftarrow \mathbf{m},
\end{aligned}
$$

where in the normal coordinate, our update is exactly the Euclidean GD with momentum.

The intermediate step in $\mathbf{B}$ allows us to include structures in $\mathbf{B}$ by considering sparse subgroup structures in $\mathbf{B}$ as suggested by Lin et al. (2021a). This gives rise to structured Gaussian (sub)manifolds as we will discuss in B. In structured cases, our update also enjoys Lie subgroup invariance (Lin et al., 2021b).

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

## Appendix A. Backgound

### A.1. Exact Riemannian exponential map and Riemannian transport map

| Manifold | Riemannian exponential map | Riemannian transport map |
|---|---|---|
| Gaussian (constant mean) (see Fig.1($a$)) | $\text{RExp}(\boldsymbol{\tau}, \mathbf{v}) = \boldsymbol{\tau}\text{Expm}(\boldsymbol{\tau}^{-1}\mathbf{v})$; symmetric | $\hat{T}_{\tau_0 \to \tau_1}(\mathbf{v}) = \mathbf{E}\mathbf{v}\mathbf{E}^T; \mathbf{E} = (\boldsymbol{\tau}_1\boldsymbol{\tau}_0^{-1})^{\frac{1}{2}}$ |
| Structured Gaussian (constant mean) | Unknown | Unknown |
| (Structured) Gaussian (see Fig.1($b$),1($c$) ) | Unknown | Unknown |

Table 1: The exact maps for the Fisher-Rao metric under global coordinate $\boldsymbol{\tau}$

### A.2. Fisher Information Matrix

Under parametrization $\boldsymbol{\tau}$, the Fisher information matrix is defined as

$$\mathbf{F}_\tau(\boldsymbol{\tau}^{(\text{cur})}) := -E_{q(z|\tau)}[\nabla_\tau^2 \log q(\mathbf{z}|\boldsymbol{\tau})], \tag{11}$$

where $q(\mathbf{z}|\boldsymbol{\tau})$ is a probabilistic distribution parameterized by $\boldsymbol{\tau}$ such as the Gaussian distribution with $\boldsymbol{\tau} = \{\boldsymbol{\mu}, \mathbf{S}\}$, where $\boldsymbol{\mu}$ is the mean and $\mathbf{S}$ is the inverse covariance matrix.

### A.3. Christoffel Symbols

The Christoffel symbols of the first kind of the Fisher metric connection are defined as

$$\Gamma_{d,ab}(\boldsymbol{\tau}) := \frac{1}{2}[\partial_a F_{bd}(\boldsymbol{\tau}) + \partial_b F_{ad}(\boldsymbol{\tau}) - \partial_d F_{ab}(\boldsymbol{\tau})]\Big|_{\tau=r}, \tag{12}$$

where $F_{bd}(\boldsymbol{\tau})$ denotes the $(b,d)$ entry of metric $\mathbf{F}_\tau$ and $\partial_b$ denotes the partial derivative w.r.t. the $b$-th entry of $\boldsymbol{\tau}$.

The Christoffel symbols of the second kind are defined as $\Gamma_{ab}^c(\boldsymbol{\tau}) := \sum_d F^{cd}(\boldsymbol{\tau})\Gamma_{d,ab}(\boldsymbol{\tau})$ where $F^{cd}(\boldsymbol{\tau})$ denotes the (c,d) entry of $\mathbf{F}_\tau^{-1}$. Observe that the Christoffel symbols involves computing all partial derivatives of the Fisher matrix $\mathbf{F}_\tau$.

### A.4. Riemannian Exponential Map

The exponential map is defined via a geodesic. The geodesic generalizes the notion of a straight line to a manifold. The geodesic $\mathbf{r}(t)$ satisifies the following second-order nonlinear system of ODEs with initial values $\mathbf{r}(0) = \mathbf{x}$ and $\dot{\mathbf{r}}(0) = \boldsymbol{\nu}$, where $\mathbf{x}$ denotes a point on the manifold and $\boldsymbol{\nu}$ is a Riemannian gradient,

$$\ddot{r}^c(t) + \sum_{a,b} \Gamma_{ab}^c(\mathbf{r}(t))\dot{r}^a(t)\dot{r}^b(t) = 0, \tag{13}$$

where $r^c(t)$ denotes the $c$-th entry and $\Gamma_{ab}^c$ is the Christoffel symbol of the second kind.

The Riemannian exponential map is defined as

$$\text{RExp}(\mathbf{x}, \boldsymbol{\nu}) := \mathbf{r}(1), \tag{14}$$

where $\mathbf{x}$ denotes an initial point and $\boldsymbol{\nu}$ is an initial Riemannian gradient so that $\mathbf{r}(0) = \mathbf{x}$ and $\dot{\mathbf{r}}(0) = \boldsymbol{\nu}$.

### A.5. Parallel Transport Map for Riemannian Gradients

In a curved space, the transport map along a given curve generalizes the notion of the parallel transport. In Riemannian optimization, we consider the transport map along a geodesic curve. Given a geodesic curve $\mathbf{r}(t)$, denote by $\mathbf{v}(t)$ a smooth Riemannian gradient field that satisfies the following first-order linear system of ODEs with initial value $\mathbf{v}(0) = \boldsymbol{\nu}$,

$$\dot{v}^c(t) + \sum_{a,b} \Gamma^c_{ab}(\mathbf{r}(t)) v^a(t) \dot{r}^b(t) = 0. \tag{15}$$

The transport map $\hat{T}_{\tau^{(\text{old})} \to \tau^{(\text{old})}}(\boldsymbol{\nu})$ transports the Riemannian gradient $\boldsymbol{\nu}$ at $\boldsymbol{\tau}^{(\text{old})}$ to $\boldsymbol{\tau}^{(\text{new})}$ as follows,

$$\hat{T}_{\tau^{(\text{cur})} \to \tau^{(\text{new})}}(\boldsymbol{\nu}) := \mathbf{v}(1), \tag{16}$$

where $\mathbf{r}(0) = \boldsymbol{\tau}^{(\text{cur})}$, $\mathbf{r}(1) = \boldsymbol{\tau}^{(\text{new})}$, and $\mathbf{v}(0) = \boldsymbol{\nu}$. It can be computationally challenging to solve this linear ODE due to the Christoffel symbol.

### A.6. Parallel Transport Map for Euclidean Gradients

Given a geodesic curve $\mathbf{r}(t)$, denote by $\boldsymbol{\omega}(t)$ a smooth Euclidean gradient field on manifold $\mathcal{M}$ that satisfies the following first-order linear system of ODEs with initial value $\boldsymbol{\omega}(0) = \mathbf{m}$,

$$\dot{\omega}_c(t) - \sum_{a,b} \Gamma^a_{cb}(\mathbf{r}(t)) \omega_a(t) \dot{r}^b(t) = 0. \tag{17}$$

The transport map transports the Euclidean gradient $\mathbf{g}$ at $\boldsymbol{\tau}^{(\text{cur})}$ to $\boldsymbol{\tau}^{(\text{new})}$ as shown below,

$$T_{\tau^{(\text{cur})} \to \tau^{(\text{new})}}(\mathbf{g}) := \boldsymbol{\omega}(1), \tag{18}$$

where $\mathbf{r}(0) = \boldsymbol{\tau}^{(\text{cur})}$, $\mathbf{r}(1) = \boldsymbol{\tau}^{(\text{new})}$, and $\boldsymbol{\omega}(0) = \mathbf{g}$.

These two transport maps are related as follows,

$$\hat{T}_{\tau^{(\text{cur})} \to \tau^{(\text{new})}}(\mathbf{F}_\tau^{-1}(\boldsymbol{\tau}^{(\text{cur})})\mathbf{g}) = \mathbf{F}_\tau^{-1}(\boldsymbol{\tau}^{(\text{new})}) T_{\tau^{(\text{cur})} \to \tau^{(\text{new})}}(\mathbf{g}), \tag{19}$$

where $\mathbf{g}$ is a Euclidean gradient evaluated at $\boldsymbol{\tau}^{(\text{cur})}$.

### A.7. A First-order Approximation of the Transport Map

We can also consider a first-order approximation of the transport map for Euclidean gradients as follows,

$$T_{\eta_0 \to \eta_1}(\mathbf{m}^{(\eta_0)}) = \boldsymbol{\omega}(1) \approx \boldsymbol{\omega}(0) + \dot{\boldsymbol{\omega}}(0), \tag{20}$$

where we have to evaluate the Christoffel symbols as discussed below.

By the transport ODE in Eq. 17, we can compute $\dot{\boldsymbol{\omega}}(0)$ as follows,

$$\dot{\omega}_c(0) - \sum_{a,b} \Gamma^a_{cb}(\mathbf{r}(0)) \omega_a(0) \dot{r}^b(0) = 0,$$

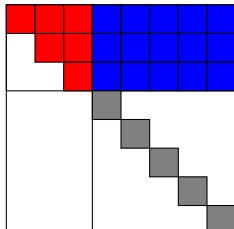

Figure 2: Visualization of the subgroup

where $\mathbf{r}(0) = \boldsymbol{\eta}_0$ is the current point and $\dot{\mathbf{r}}(0)$ is the Riemannian gradient so that $\boldsymbol{\eta}_1 = \text{Exp}(\boldsymbol{\eta}_0, \dot{\mathbf{r}}(0))$. In our case, $\dot{\mathbf{r}}(0) = -\mathbf{F}_\eta^{-1}(\eta_0)\mathbf{m}^{(\eta_0)}$ and $\boldsymbol{\omega}(0) = \mathbf{m}^{(\eta_0)}$.

Note that the Fisher matrix and Christoffel symbols are evaluated at a generalized Riemannian normal coordinate $\boldsymbol{\eta}_0 = \mathbf{0}$. The computation can be simplified due to the identities: $\mathbf{F}_\eta^{-1}(\boldsymbol{\eta}_0) = \mathbf{I}$ and $\Gamma_{cb}^a(\boldsymbol{\eta}_0) = \sum_d F^{ad}(\boldsymbol{\eta}_0)\Gamma_{d,cb}(\boldsymbol{\eta}_0) = \sum_d \delta^{ad}\frac{1}{2}[\partial_c F_{bd}(\boldsymbol{\eta}_0) + \partial_b F_{cd}(\boldsymbol{\eta}_0) - \partial_d F_{cb}(\boldsymbol{\eta}_0)]$, where $F^{ad}(\boldsymbol{\eta}_0) = \delta^{ad}$. Thus, we have the following simplification.

$$\dot{\omega}_c(0) = -\frac{1}{2}\sum_{b,d}[\partial_c F_{bd}(\boldsymbol{\eta}_0) + \partial_b F_{cd}(\boldsymbol{\eta}_0) - \partial_d F_{cb}(\boldsymbol{\eta}_0)](\mathbf{m}^{(\eta_0)})^d(\mathbf{m}^{(\eta_0)})^b$$

$$= -\frac{1}{2}\sum_{b,d}\partial_c F_{bd}(\boldsymbol{\eta}_0)(\mathbf{m}^{(\eta_0)})^d(\mathbf{m}^{(\eta_0)})^b$$

In the Gaussian case considered in Sec. 3, we can show the first-order term vanishes since $\mathbf{m}$ is symmetric.

$$\dot{\omega}(0) = -\frac{1}{2}\Big(\frac{1}{\sqrt{2}}\Big[\big(\frac{1}{\sqrt{2}}\mathbf{m}\big)\big(\frac{1}{\sqrt{2}}\mathbf{m}\big)^T - \big(\frac{1}{\sqrt{2}}\mathbf{m}\big)^T\big(\frac{1}{\sqrt{2}}\mathbf{m}\big)\Big]\Big) = -\frac{1}{2\sqrt{2}}\Big( < \frac{1}{\sqrt{2}}\mathbf{m}, (\frac{1}{\sqrt{2}}\mathbf{m})^T > \Big) = 0,$$

where $< \mathbf{m}, \mathbf{m}^T > := \mathbf{m}\mathbf{m}^T - \mathbf{m}^T\mathbf{m}$ is the Lie bracket.

### A.8. Update 5 is Equivalent to Alimisis et al. (2020)

The update of Alimisis et al. (2020) is defined as

$$\boldsymbol{\nu}^{(\text{cur})} \leftarrow \beta\mathbf{z}^{(\text{cur})} + \alpha\mathbf{F}_\tau^{-1}(\boldsymbol{\tau}^{(\text{cur})})\mathbf{g}(\boldsymbol{\tau}^{(\text{cur})}),$$
$$\boldsymbol{\tau}^{(\text{new})} \leftarrow \text{RExp}(\boldsymbol{\tau}^{(\text{cur})}, -\boldsymbol{\nu}^{(\text{cur})}),$$
$$\mathbf{z}^{(\text{new})} \leftarrow \hat{T}_{\tau^{(\text{cur})}\to\tau^{(\text{new})}}(\boldsymbol{\nu}^{(\text{cur})}), \tag{21}$$

where $\boldsymbol{\nu}$ is a Riemannian gradient, and $\hat{T}_{\tau^{(\text{cur})}\to\tau^{(\text{new})}}$ is the parallel transport map for Riemannian gradients.

Due to Eq. 19, update 21 and update 5 are equivalent by the observation that $\mathbf{m}^{(\text{cur})} = \mathbf{F}(\boldsymbol{\tau}^{(\text{cur})})\boldsymbol{\nu}^{(\text{cur})}$ and $\mathbf{w}^{(\text{new})} = \mathbf{F}(\boldsymbol{\tau}^{(\text{new})})\mathbf{z}^{(\text{new})}$.

## Appendix B. Structured Gaussian with Constant Mean

Consider a $d$-dimensional Gaussian family with zero mean $q(\mathbf{z}|\boldsymbol{\tau})$, where the global parametrization $\boldsymbol{\tau}$ is the inverse covariance, which is symmetric positive-definite.

We introduce an auxiliary parameterization $\mathbf{B}$ such that $\boldsymbol{\tau} = \mathbf{B}^T\mathbf{B}$ and $\mathbf{B} = \begin{bmatrix} \mathbf{B}_A & \mathbf{B}_C \\ \mathbf{0} & \mathbf{B}_D \end{bmatrix}$, where $\mathbf{B}_A$ is an invertible upper-triangular matrix and $\mathbf{B}_D$ is an invertible diagonal matrix. The auxiliary parameter $\mathbf{B}$ lives in a subgroup of the upper-triangular group illustrated in Figure 2.

We first define a constant matrix $\mathbf{D} = \begin{bmatrix} \frac{\mathrm{Tril}(\mathbf{1}_A)}{\sqrt{2}} & \mathbf{1}_B \\ \mathbf{0} & \frac{1}{\sqrt{2}}\mathbf{I}_D \end{bmatrix}$, where $\mathbf{1}$ is the matrix of all ones, $\mathbf{I}_D$ is the identity matrix, $\mathrm{Tril}()$ extracts the upper-triangular half. We define a local parameterization $\boldsymbol{\eta} = \begin{bmatrix} \boldsymbol{\eta}_A & \boldsymbol{\eta}_C \\ \mathbf{0} & \boldsymbol{\eta}_D \end{bmatrix}$ associated to $\mathbf{B}^{(\mathrm{cur})}$ as $\mathbf{B} = \mathbf{h}(\mathbf{D}\odot\boldsymbol{\eta})\mathbf{B}^{(\mathrm{cur})}$, where $\odot$ is the element-wise product, $\mathbf{h}(\mathbf{N})$ is the matrix exponential, $\boldsymbol{\eta}_A$ is an upper-triangular matrix, $\boldsymbol{\eta}_D$ is a diagonal matrix. In this case, the local parameter $\boldsymbol{\eta}$ lives in the Lie algebra of $\mathbf{B}$.

We can show that $\boldsymbol{\eta}$ is a generalized Riemannian normal coordinate associated to $\boldsymbol{\tau}^{(\mathrm{cur})}$ as discussed in the main text. Moreover, $\mathbf{F}_\eta(\boldsymbol{\eta}_0) = \mathbf{I}$ where $\boldsymbol{\eta}_0 = \mathbf{0}$. Since $\mathbf{m}$ has the same structure of $\boldsymbol{\eta}$, $\mathbf{m}$ is not symmetric. The first-order approximation of the transport map at $\boldsymbol{\eta}_0$ do not vanish since $\mathbf{m}$ is not symmetric. However, in practice, the first-order approximation of the transport does not improve the performance since step-size $\alpha$ is small. Recall that $\mathbf{m}$ is $O(\alpha)$ in our proposed update. The first-order approximation is $O(\alpha^2)$ as shown below.

$$\dot{\boldsymbol{\omega}}(0) = -\tfrac{1}{2}(\mathbf{D}\odot((\mathbf{D}\odot\mathbf{m})(\mathbf{D}\odot\mathbf{m})^T - (\mathbf{D}\odot\mathbf{m})^T(\mathbf{D}\odot\mathbf{m}))) = -\tfrac{1}{2}\underbrace{(\mathbf{D}\odot <\mathbf{D}\odot\mathbf{m}, (\mathbf{D}\odot\mathbf{m})^T>)}_{O(\alpha^2)},$$

where $<A, B> := AB - BA$ is the Lie bracket.

Notice that $\boldsymbol{\tau} = (\mathbf{B}^{(\mathrm{cur})})^T\mathbf{h}^T(\mathbf{D}\odot\boldsymbol{\eta})\mathbf{h}(\mathbf{D}\odot\boldsymbol{\eta})\mathbf{B}^{(\mathrm{cur})} = (\mathbf{B}^{(\mathrm{new})})^T\mathbf{h}^T(\mathbf{D}\odot\boldsymbol{\xi})\mathbf{h}(\mathbf{D}\odot\boldsymbol{\xi})\mathbf{B}^{(\mathrm{new})}$ and $\mathbf{B}^{(\mathrm{new})} = \mathbf{h}(\mathbf{D}\odot\boldsymbol{\eta}_1)\mathbf{B}^{(\mathrm{cur})}$. We have $\mathbf{h}^T(\mathbf{D}\odot\boldsymbol{\eta})\mathbf{h}(\mathbf{D}\odot\boldsymbol{\eta}) = \mathbf{h}^T(\mathbf{D}\odot\boldsymbol{\eta}_1)\mathbf{h}^T(\mathbf{D}\odot\boldsymbol{\xi})\mathbf{h}(\mathbf{D}\odot\boldsymbol{\xi})\mathbf{h}(\mathbf{D}\odot\boldsymbol{\eta}_1)$. Due to the structure restriction on $\boldsymbol{\eta}$ and $\boldsymbol{\xi}$, $\mathbf{h}(\mathbf{D}\odot\boldsymbol{\eta})$ and $\mathbf{h}(\mathbf{D}\odot\boldsymbol{\xi})$ cannot be an orthogonal matrix and change the sign of the determinant. Thus, $\mathbf{h}(\mathbf{D}\odot\boldsymbol{\eta}) = \mathbf{h}(\mathbf{D}\odot\boldsymbol{\xi})\mathbf{h}(\mathbf{D}\odot\boldsymbol{\eta}_1)$. Since $\mathbf{h}$ is the matrix exponential, we have $\boldsymbol{\eta} = \mathbf{D}^{-1}\odot\mathrm{Expm}^{-1}\big(\mathrm{Expm}(\mathbf{D}\odot\boldsymbol{\xi})\mathrm{Expm}(\mathbf{D}\odot\boldsymbol{\eta}_1)\big)$.

By the Baker–Campbell–Hausdorff formula, we have $\boldsymbol{\eta} = \boldsymbol{\xi} + \boldsymbol{\eta}_1 + (2\mathbf{D})^{-1}\odot <\mathbf{D}\odot\boldsymbol{\xi}, \mathbf{D}\odot\boldsymbol{\eta}_1> +O(\boldsymbol{\xi}^2) + O(<\boldsymbol{\eta}_1, <\boldsymbol{\eta}_1, \boldsymbol{\xi}>)$ , where $<A, B> := \mathbf{AB} - \mathbf{BA}$ is the Lie bracket. We consider the zero-order approximation of the transport map. In this case, $\mathbf{w} = \mathbf{m}$ and $\boldsymbol{\eta}_1 = -\mathbf{m}$. Recall that $\mathbf{m} = O(\alpha)$, where $\alpha$ is the step size. The Jacobian-vector product at $\boldsymbol{\xi}_0 = \mathbf{0}$ can be simplified as

$$\mathbf{J}(\boldsymbol{\xi}_0)\mathbf{w} = \mathbf{J}(\boldsymbol{\xi}_0)\mathbf{m}$$
$$= \frac{\partial\mathrm{Tr}(\mathbf{m}^T\boldsymbol{\eta})}{\partial\boldsymbol{\xi}}\Big|_{\boldsymbol{\xi}=\boldsymbol{\xi}_0} = \underbrace{\mathbf{m}}_{O(\alpha)} + \underbrace{\frac{\mathbf{D}}{2}\odot <\mathbf{D}^{-1}\odot\mathbf{m}, (\mathbf{D}\odot -\mathbf{m})^T>}_{=O(\alpha^2)} + \underbrace{O(<\mathbf{m}, <\mathbf{m}, \mathbf{m}^T>>)}_{=O(\alpha^3)}$$

In the Gaussian with constant mean cases, the high-order terms in the Jacobian-vector product all vanish since $\mathbf{m}$ is symmetric and $\mathbf{D} = \frac{1}{\sqrt{2}}\mathbf{1}$, where $\mathbf{1}$ is a matrix of all ones. In structured Gaussia cases, the high-order terms in the Jacobian-vector product do not vanish. However, in practice, we can ignore the high-order terms since the step-size $\alpha$ is very small and the high-order terms do not improve performance. We can similarly obtain the proposed update.

## Appendix C. Numerical Results

### C.1. Optimization on Symmetric Positive-definite Matrices

We consider manifold optimization on positive-definite matrices $\mathcal{S}_{++}^{d \times d}$, where the Riemannian exponential and transport maps admit a closed-form expression as shown in Table.1. However, our approach can be extended to structured cases, where these maps do not have a closed-form expression.

We evaluate the proposed method in an example of the metric nearness problem considered in Lin et al. (2021a). The objective function is $\ell(\mathbf{Z}) = \frac{1}{2N} \sum_{i=1}^{N} \|\mathbf{Z}\mathbf{Q}\mathbf{x}_i - \mathbf{x}_i\|_2^2$, where $\mathbf{x}_i \in \mathcal{R}^d$, $\mathbf{Q} \in \mathcal{S}_{++}^{d \times d}$ and $\mathbf{Z} \in \mathcal{S}_{++}^{d \times d}$. The optimal is $\mathbf{Q}^{-1}$. We randomly generate $\mathbf{x}_i$ and $\mathbf{Q}$ with $d = 100$, $N_{\text{train}} = 5 \times 10^5$ for training and $N_{\text{test}} = 10^5$ for testing. All methods are trained using mini-batches, where the size of the mini-batch is 250. We also consider existing Riemannian momentum methods such as Ahn and Sra (2020), Alimisis et al. (2021), and Alimisis et al. (2020) as baselines. From Figure 1(a), we can see that the proposed method using a local coordinate performs as well as existing Riemannian momentum methods using a global coordinate with the exact exponential and transport maps.

### C.2. Unconstrained Optimization

We consider the following valley-shaped function suggested by in Lin et al. (2021a). Rosenbrock : $\ell(\mathbf{z}) = \frac{1}{d} \sum_{i=1}^{d-1} \left[ 100(z_{i+1} - z_i)^2 + (z_i - 1)^2 \right]$, where $d = 10^3$. In this example, we consider a structured Gaussian manifold with a non-constant mean, where these exponential and transport maps do not have a closed-form expression. Our baselines are the BFGS provided by SciPy and the method of Lin et al. (2021a). Our method and the method of Lin et al. (2021a) use a structured Hessian information, which can be efficiently computed via a Hessian-vector product. We consider a block Heisenberg structure as suggested by Lin et al. (2021a), where our update introduce momentum into the method of Lin et al. (2021a). From Figure 1(b), we can see that the momentum can improve the performance.

### C.3. Deep Learning

We consider a CNN model with the VGG-16 neural network architecture. We train the model with matrix Gaussian (Lin et al., 2021a) for each layer-wise matrix weights on dataset "CIFAR-10". In this example, we consider a structured Gaussian manifold with a non-constant mean, where these exponential and transport maps do not have a closed-form expression. For each Gaussian, we use a Kronecker product of two dense groups. The iteration cost and the structure are similar to the KFAC approximation (Martens and Grosse, 2015). We also use a similar approximation as KFAC to compute natural gradients. We include the KFAC method as our baseline method. We train the model with mini-batch size 256. From Figure 1(c), we can see that the momentum can improve the performance and our method performs similarly to KFAC.

However, our approach enables us to introduce more sparse structures along with the Kronecker structure such as a Kronecker product of two sparse groups, which can further reduce the iteration cost of our method. We are currently investigating sparse updates that can further reduce the iteration cost while maintaining the performance.

