# OpenReview forum: "Practical Structured Riemannian Optimization with Momentum by using Generalized Normal Coordinates"
_NeurIPS.cc/2022/Workshop/NeurReps — NeurReps 2022 Poster_

### Official Review · Reviewer_49mX · 2022-10-10
**Review of Riemannian Momentum**

**Confidence:** 4
**Soundness:** 4
**Presentation:** 3
**Contribution:** 4
**Overall Rating:** 7

**Summary:**

The authors propose a method to perform optimization on Riemannian manifolds with momentum. The authors introduce a generalization of the Riemannian momentum update to allow for other parameterizations (rather than just the exponential map). They then use an approximation of the corresponding parallel transport map which does not require Christoffel symbols. The proposed method is analyzed for the case of Gaussian Fisher-Rao manifolds, where they let $\tau = B^\top B$ and $B' = \exp(\eta / \sqrt{2} ) B$. The authors compare their method with other Riemannian optimization approaches on a manifold optimization problem and also provide comparison to alternative deep network optimizers for a CIFAR-10 classification problem.

**Questions:**

Do the authors have an example where they can compute the Christoffel symbols so that the approximation can be compared to the exact update in equation 5?

Also, if the authors could briefly comment on the comparison of this work to Lin et al and Alimesis et al, it would be appreciated.

Lastly, how large are the numerical efficiency gains associated with approximating the transport map?

**Limitations:**

The authors adequately addressed limitations of their work.

**Recommended Decision:**

3: Accept

**Relevance:**

4: Highly relevant

**Strengths And Weaknesses:**

Strengths

The paper gives a novel approach to momentum based training in Riemannian optimization. The generalization to alternative parameterization systems is quite interesting, and the subsequent approximations of the transport maps appear to give good performance.

The paper includes compelling numerical experiments for a metric nearness problem, the Rosenbrock problem, and a computer vision task, showing that their method is competitive with other candidate methods.

Weakness

One concern is the relative novelty. In particular, how different is the update proposed in this work and the previous works of Lin et al and Alimesis et al? In A.7 the authors claim that their update 5 matches the update in Alimesis et al. Is the main difference the approximation scheme?

The other weakness of the paper is that many of the important concepts are only defined in the appendix. For a reader unfamiliar with Riemannian geometry and transport maps, a few sentences defining these ideas in the main text could be helpful.

**Submission Track:**

Extended Abstract (4 Page)

---

### Official Review · Reviewer_finE · 2022-10-13
**Interesting topic broadly**

**Confidence:** 3
**Soundness:** 3
**Presentation:** 3
**Contribution:** 3
**Overall Rating:** 5

**Summary:**

This abstract describes some variations on Riemannian gradient descent that incorporate momentum.  If I understand properly, these methods are intended to be easier to work with because they need a weaker notion of normal coordinates than past alternatives.

**Questions:**

Maybe add a reference for momentum in first paragraph of section 1

Define NGD acronym

Sec 1.2, paragraph 2, “constants” --- These are only constants if tau^(cur) is held constant, right?

Eq 3:  Why define eta_0=0 and then use eta_0 instead of just using 0 everywhere? This seems like unnecessary notation.  Similarly in (4) it seems like eta_0 doesn’t add much, but maybe I’m misunderstanding the message here.

It seems like the first paragraph of sec 2.1 is missing references to past work.  Is this section (and in particular eq5) entirely new?  If not set the stage more carefully here.  What already existed, and what’s your contribution relative to state of the art?

It would be useful to provide some intuition in sec 2.2 for the generalized parametrization.  How often is such a parametrization available for a given manifold?  Are there examples where it’s easier to find one of these than the usual normal coordinates?

It seems like (7) is an aggressive approximation.  Doesn’t this sort of remove the geometric structure you’re trying to leverage?  If I understand correctly, parallel transport is basically ignored.

I wasn’t sure the role of section 3.  What’s the difference between the algorithm defined here and what came in previous sections?  Why is section 3 needed?


**Limitations:**

Not too many limitations are discussed.  I would appreciate more intuitive discussion of the generalized coordinates (e.g. some practical examples), including when we expect them to be possible to obtain.

**Recommended Decision:**

3: Accept

**Relevance:**

3: Solid fit

**Strengths And Weaknesses:**

The broad topic area of the paper is interesting, and the setting of the paper was straightforward to read and understand.  I like the overall problem the authors are tacking, and the (very) few experiments here seem to show a little value of the proposed algorithm.

The exposition in the second half of this extended abstract became quite hard to follow; I think the authors tried to compress too much into four pages and ended up making the work hard to follow.  Section 3 in particular doesn't seem motivated.

I'm also concerned that approximation (7) seems to remove all the geometry from the algorithm.

**Submission Track:**

Extended Abstract (4 Page)

---

### Official Review · Reviewer_hNdC · 2022-10-14
**New method with good results. Writing could be improved upon.**

**Confidence:** 2
**Soundness:** 3
**Presentation:** 1
**Contribution:** 3
**Overall Rating:** 7

**Summary:**

The authors propose a new way to perform optimization on manifolds by 1) presenting new local coordinates that simplify calculations and 2) introduce momentum into their local coordinate calculations. Authors show that their local coordinates without momentum perform on par with other standard methods which use global coordinates, and they show that their local coordinates with momentum improves upon the performance.

**Questions:**

- The authors never define what they mean by “momentum”. I know of the physics definition of momentum (p=mv, etc.), but I do not know if this is the momentum they are talking about. If it is the momentum they are talking about, I don’t understand how momentum is used in manifold optimization problems. If the physics definition is not the momentum they are talking about, then I don’t know what momentum they ARE referring to. Since “momentum” seems to be central to their paper, I think it would have been helpful for readers if they had spent some time discussing what it is.
- Authors say they build on the structured NGD method introduced by Lin et al. (2021 a), but they never define the acronym NGD, and they never describe what this method is. Therefore, I have no idea what method they are trying to build off of. Since the “structured NGD method” seems to be central to their paper, I think they should have spent some time explaining what it is.
-There is no description of figure 1 in the main text. Authors refer readers to the appendix for a description of their only figure, but I think that the main text should be self contained and not rely on information that is contained in the appendix. In the main text, they do not have any description of their only figure.
- Figure 1 title "Preliminary Numerical Results” is extremely vague. Figures generally should be self contained (readers should be able to look at a figure and understand what it is showing/why it is important without reading the text around it). For example, I did not understand that the numerical results shown in these figures included momentum until looking at the appendix because 1) it does not say so and 2) the figure is BEFORE the momentum part of the paper.
- Figure 1 is not referenced in the main text (i.e. there is no sentence in the main text that says something like “… as shown in figure 1” or “see figure 1”). Figures should be referenced in main text.


**Limitations:**

I think these are exciting results! My main comments for improvement are for improving the clarity of the results and the clarity of the problem statement (what you are doing).

**Recommended Decision:**

3: Accept

**Relevance:**

3: Solid fit

**Strengths And Weaknesses:**

Originality:
The authors build on Lin et al. with their own original idea for local coordinates. They improve on their local coordinate results by incorporating momentum, which is commonly used.

Quality:
The numerical results of their method outperforms other common methods.

Clarity:
The clarity of the submission could be improved. See “questions” section. Overall, I would suggest they 1) fully define the most important methods in their paper 2) make the paper self-contained (so that readers do not need to read the appendix to understand it) 3) make the figure self-contained

Significance:
Because this method outperforms many other methods, these results are of interest to the community, since optimization on manifolds is of interest to the community.


**Submission Track:**

Extended Abstract (4 Page)

---

### Author Response · Authors · 2022-11-02
**Author response (to all reviewers)**

We thank the reviewers for taking the time to read and comment on this abstract.

This abstract aims to discuss key techniques of an ongoing project and machine learning applications in an informal format.

Due to the 4-page limit of the original submission, it is impossible for us to include a thorough literature review, detailed background materials, and in-depth technical analysis in the main text.

Below we discuss the common issues brought up in the reviews.


* **Presentation**:
We agree that the presentation can be improved.

We made the following major changes in the camera-ready version since we have an extra page.
1. A short discussion about related works is included in Sec 2.1.
2. We focus on Gaussian Fisher-Rao manifolds, which have a Lie-group structure, and their important roles in machine learning.
3. A short discussion about the zero-order approximation is included in Sec 2.4.
4. More Technical discussions about approximations made in our method are included in appendices.

---
* **Originality/Contributions**:
It is challenging to introduce momentum along a Riemannian metric under local coordinates in a computationally efficient way. Since a local coordinate will change at each iteration, adding momentum in local coordinates will involve the computation of the Riemannian metric, the exponential map, the transport map, and the Jacobian matrix.

 We made the following contributions:
1. We propose to use generalized normal (local) coordinates, which enables us to include momentum and large-scale usage of our updates with a low computation cost.
2. To our knowledge, Eq 5 is new, which connects our approach to existing Riemannian momentum methods.  In Eq 5, we propose to use a new transport map,  which can be efficiently approximated later.
3. We show that Eq 5 (in a metric-compatible way) can be viewed as Euclidean GD with momentum in a standard Riemannian normal coordinate. We argue that a generalized normal coordinate can replace the standard normal coordinate in Gaussian Fisher-Rao manifolds.

---
* **Comparison to Lin et al 2021**:
Lin et al 2021 propose an update scheme using local coordinates for Gaussian Fisher-Rao manifolds. However, It is non-trivial to include momentum in the original scheme due to the change between two local coordinates at each iteration.  Adding momentum in local coordinates will involve the computation of the Riemannian metric, the exponential map, the transport map, and the Jacobian matrix. We show that our proposed normal coordinates can simplify the computation while the original local coordinates proposed by Lin et al 2021 do not.

---
* **Comparison to Alimisis et al 2020**:
To the best of our best knowledge, many existing Riemannian methods including Alimisis et al 2020 work on a global coordinate and use the exact Riemannian exponential and the exact Riemannian transport map. These maps are unknown especially in structured Gaussian (sub-manifold) cases. Moreover, these maps can be computationally intensive to approximate due to the computation of Christoffel symbols under the global coordinate.

---

### Author Response · Authors · 2022-11-02
**Author response (to reviewer hNdC)**

* **Presentation**: see the response to all reviewers.
We made several changes to improve the presentation such as updating the caption of Figure 1 in the camera-ready version.
Note that we cannot include detailed background materials and self-contained numerical results in the main text due to the page limit.

---

### Author Response · Authors · 2022-11-02
**Author response (to reviewer finE)**

* **Contributions/Novelty**: see the response to all reviewers

---
* **The role of Section 3 about Gaussian Fisher-Rao manifolds**:
> Section 3 in particular doesn't seem motivated

The role of Section 3 is to illustrate our proposed method and the approximation (Eq 7)  in a concrete example. In the original submission, we did mention this point in the introduction.
In the camera-ready version, we explicitly emphasize the importance of Gaussian Fisher-Rao manifolds in machine learning.  Moreover, Gaussian Fisher-Rao manifolds have an additional Lie-group structure.  We also suggest a normal coordinate in Sec 3 by leveraging the Lie-group structure.

---
* **More discussions about approximation (7)**
> parallel transport is basically ignored

Sec 2.4, Appendix A.6, and B are updated to address this point. Thanks to our normal coordinate,  we can explicitly compute the first-order term and show that the first-order term is $O(\alpha^2)$, where $\alpha$ is the step size.  Our point is that under our local coordinate, the parallel transport is locally like the transport in the Euclidean/flat case when the step size is small.
In practice, especially in noisy, large-scale, single-precision floating-point settings, the step size $\alpha$ has to be very small, say, $10^{-5}$. Thus, it is reasonable to ignore the first-order term. Note that there is a trade-off between the iteration/computation cost and the approximation accuracy of the transport map in practice.

---
* **The contribution of Eq 5 in Sec 2.1** :

To our knowledge, Eq 5 is new, which connects our approach to existing Riemannian momentum methods.  In Eq 5, we propose to use a new transport map,  which can be efficiently approximated later.  We show that Eq 5 (in a metric-compatible way) can be viewed as Euclidean GD with momentum in a standard Riemannian normal coordinate. We argue that a generalized normal coordinate can replace the standard normal coordinate in Gaussian Fisher-Rao manifolds.

---
* **Intuition and examples of the generalized normal coordinate in Sec 2.2**:

The intuition of the generalized coordinate comes from natural-gradient descent. Note that natural-gradient descent is a practical approximation of Riemannian gradient descent. Likewise, our generalized normal coordinate is a practical approximation of the standard normal coordinate.
We could construct a generalized normal coordinate by a Taylor approximation of the Riemannian exponential map. Note that a Taylor approximation of the exponential map is much easier to compute than the exponential map since the Taylor approximation is a point approximation while the exponential map is the solution of the geodesic ODE. In Sec 3, we also give a normal coordinate by leveraging the Lie-group structure.

---
* **Notation of the local coordinate $\eta_0=0$**:
> Why define eta_0=0 and then use eta_0 instead of just using 0 everywhere?

Note that a local coordinate and the corresponding origin are associated with the current point/iteration. The origin ($\eta_0=0$) at the current iteration is different from the origin ($\epsilon_0=0$) at the next iteration as discussed in Sec 2.3. We have to explicitly distinguish these two origins.  The main challenge of adding momentum in local coordinates comes from the change of coordinate systems at each iteration, which involves the computation of the Riemannian metric, the exponential map, the transport map, and the Jacobian matrix.

---

### Author Response · Authors · 2022-11-02
**Author response (to reviewer 49mX)**



* **Novelty**: see the response to all reviewers

* **Comparison to Lin et al 2021**:
Lin et al 2021 propose an update scheme using local coordinates for Gaussian Fisher-Rao manifolds. However, It is non-trivial to include momentum in the original scheme due to the change between two local coordinates at each iteration.  Adding momentum in local coordinates will involve the computation of the Riemannian metric, the exponential map, the transport map, and the Jacobian matrix. We show that our proposed normal coordinates can simplify the computation while the original local coordinates proposed by Lin et al 2021 do not.


* **Comparison to Alimisis et al 2020**:
To the best of our best knowledge, many existing Riemannian methods including Alimisis et al 2020 work on a global coordinate and use the exact Riemannian exponential and the exact Riemannian transport map. These maps are unknown especially in structured Gaussian (sub-manifold) cases. Moreover, these maps can be computationally intensive to approximate due to the computation of Christoffel symbols under the global coordinate.

* **Examples of the computation of Christoffel symbols** can be found in appendix A.6 and B.

* **Numerical efficiency gains in approximating the transport map**:
There is a trade-off between the iteration/computation cost and the approximation accuracy of the transport map.
Thanks to our normal coordinate, the zero-order approximation is often good enough since the first-order term is $O(\alpha^2)$ as shown in Appendix A.6 and B, where $\alpha$ is the step size.
In practice, especially in noisy, large-scale, single-precision floating-point settings, the step size $\alpha$ has to be very small, say, $10^{-5}$.  According to our experiments, the first order term does improve the performance. Thus, we keep the iteration cost low by ignoring the insignificant first-order term.

---

### Decision · Program_Chairs · 2022-10-21

Accept (Poster)